# OVERCOMING GENERIC KNOWLEDGE LOSS WITH SELECTIVE PARAMETER UPDATE

## ABSTRACT

Foundation models encompass an extensive knowledge base and offer remarkable transferability. However, this knowledge becomes outdated or insufficient over time. The challenge lies in continuously updating foundation models to accommodate novel information while retaining their original capabilities. Leveraging the fact that foundation models have initial knowledge on various tasks and domains, we propose a novel approach that, instead of updating all parameters equally, localizes the updates to a sparse set of parameters relevant to the task being learned. We strike a balance between efficiency and new tasks performance, while maintaining the transferability and generalizability of foundation models. We extensively evaluate our method on foundational vision-language models with a diverse spectrum of continual learning tasks. Our method achieves improvements on the newly learned tasks accuracy up to 7% while preserving the pretraining knowledge with a negligible decrease of 0.9% on a representative control set accuracy.

## 1 INTRODUCTION

Recent machine learning models trained on a broad dataset have shown remarkable success in both natural language processing tasks (OpenAI, 2023) and computer vision tasks (Radford et al., 2021; Alayrac et al., 2022). These models can directly solve a wide range of tasks, such as recognizing common objects and answering common questions, thus are dubbed as foundation models (Bommasani et al., 2021). What is captured by these models covering various domains and tasks can be referred to as generic knowledge. Despite this, foundation models could still perform poorly on specific tasks. For instance, Xiang et al. (2023) found ChatGPT is limited in embodied tasks, while CLIP (Radford et al., 2021) is shown struggling in recognizing fine-grained classes like cars from different brands. Therefore, it is crucial to integrate newly revealed data with pre-trained foundation models and expand their knowledge base. As one common solution, finetuning foundation models on new data would usually result in a good performance on the new task if done carefully. This will turn the foundation model into a specific model for a specific task, and would risk losing the existing capabilities of the model or the generic knowledge it has acquired through long phases of pretraining. The effect of deteriorating the model's previous knowledge upon new learning is a typical phenomenon of neural networks, referred to as catastrophic forgetting (McCloskey & Cohen, 1989).

Continual learning research has been exploring the problem of accumulating knowledge without forgetting Parisi et al. (2019) over the past years and has provided valuable techniques. However, most existing works consider this process starting from a randomly initialized model (Ebrahimi et al., 2019; 2020). Recently, with the success of large pretrained models (Steiner et al., 2021; Wightman, 2019), many works have considered continual learning starting from a pre-trained model (Wang et al., 2022c;b). Nevertheless, the emphasis lies mostly on the learning and forgetting behavior of the newly acquired knowledge, in the upcoming task sequence, often side-lining the pre-trained knowledge. Generic knowledge embedded in large models provides bases for strong performance in various domains and quick transfer to different tasks; when continuously finetuning a large pretrained model on newly received tasks with no regard to preserving its pre-existing knowledge, we are losing the pre-training benefits and being left with merely a large model to deal with.

These prompt a crucial question: Can we effectively and continuously update foundation models while retaining their generic knowledge? An example is accommodating a generic multimodal model like CLIP (Radford et al., 2021) to specific fine-grained concepts as various types of vehicles while

maintaining its generic recognition capabilities of common world concepts such as people, animals, and plants. In a continuously evolving world, we need to design models to cope with the change.

A particular avenue of recent approaches advocates for preserving the knowledge while refining foundation models through model editing techniques (Meng et al., 2022; Mitchell et al., 2022a;b). The key is to identify specific layers to modify and perform local updates to pre-trained models, thereby correcting the related concepts without hurting other knowledge. While these methods have shown promise in incorporating specific concepts into the model, their impact on the generic knowledge remains uncertain, as discussed by Onoe et al. (2023). Additionally, most of these techniques are designed for specific models for small-scale sample-wise edit of concrete mistakes and updates. Moreover, they are centered around language models, where the input data has a stronger relationship to the concept being edited, leaving the vision models, where the input images can contain various of unrelated visual concepts, relatively unexplored. In contrast, we are interested in allowing continuous model updates on a set of new coming data samples, which can be scaled up to a larger number of concepts and a longer never-ending sequence.

Towards this goal, we seek to update foundation models from a continual learning perspective while preserving their previously acquired generic knowledge. Starting from a large model pretrained on vast sources of data, it is reasonable to assume that the model has some kind of basic or related knowledge on the new upcoming data. Thus, we hypothesize that there is an implicit modularity in the foundation model and design a method to locate which parameters are most relevant to the new upcoming data. Formally, we first identify specific model layers to be updated based on model analysis works (Dar et al., 2022; Geva et al., 2021). Among the localized layers, we propose a mechanism to select parameters that are specialized for the task at hand. We opt for selecting parameters that small changes to their values would contribute to a greater improvement in the new task performance compared to other parameters. We localize and update only a small number of the selected model's parameters, while keeping a large portion of the model's parameters untouched. By only updating a few parameters, we not only provide an efficient method to finetune a large pretrained model on newly arriving data but also preserve greatly the generalizability and transferability of the model. Our strategy is to be executed whenever new data corresponding to a new set of classes, a new task or domain, is received.

To facilitate a comprehensive analysis of the generic knowledge deterioration, we focus on the classification tasks and formulate the knowledge base as the zero-shot classification ability on a diverse control set containing a wide range of classes. Our main objective is to demonstrate an improvement of a pre-trained model's performance on datasets where it initially exhibits suboptimal results, while preserving its original ability on a control set, without revisiting it. We evaluate our method on six continual learning tasks and find that by updating merely 3% of the parameters, our approach achieves performance on the new tasks superior to that achieved by methods that fully finetune the model, with almost no deterioration on the generic knowledge, only 0.97% performance loss on the control set. We further conduct comprehensive analyses to assess the impact of each component on generic knowledge forgetting.

Our contribution can be concluded as 1) We introduce the evaluation of generic knowledge forgetting in continual learning starting from foundation models. 2) To ensure the preservation of pre-trained knowledge, we propose an efficient method that localizes the learnable parameters, selects specialized parameters for the new coming data, and performs sparse updates. 3) Through comprehensive evaluations on six datasets, we demonstrate that our algorithm significantly expands the pre-trained knowledge on new tasks while still preserving the generic knowledge. Additionally, we conduct in-depth analyses to understand the impact of each component on generic knowledge forgetting.

## 2 RELATED WORK

**Foundation Models.** Pretraining techniques have played a crucial role in establishing the so-called foundation models, such as CLIP (Radford et al., 2021), Flamingo (Alayrac et al., 2022), BLIP-2 (Li et al., 2023), PaLM-E (Driess et al., 2023), and GPT-4 (OpenAI, 2023). These models are pre-trained on vast and diverse datasets, providing them with a broad knowledge base and exceptional generalization and transferability. Consequently, many of these models can be directly applied to various tasks in a zero-shot manner. Despite their strong abilities, evaluating these foundation models remains challenging (Xu et al., 2023), given that their strengths lie predominantly in a diverse domain of generalization. While CLIP (Radford et al., 2021), an early vision-language model pre-trained

on a large dataset of 400 million images and text samples, namely WebImageText, is an exception that exhibits impressive performance mainly on zero-shot classification tasks. This straightforward evaluation format allows us to thoroughly explore the changes in the model's knowledge base when implementing updates or modifications. By studying the impact of these changes on CLIP, we aim to gain a more in-depth understanding of the potential of updating the foundation models.

**Continual Learning.** In the realm of continual learning, early methods (Chaudhry et al., 2019b; Kirkpatrick et al., 2017; Chaudhry et al., 2019a; Ebrahimi et al., 2020) train models from scratch for each specific sequence. Recent methods leverage the power of pre-trained models to handle a new sequence of tasks. Piggyback (Mallya et al., 2018), as a pioneer, learns separate masks over a frozen pre-trained model for different tasks in the sequence. It requires storing the masks and access to task identification to apply the mask during inference, which is a limiting assumption. Another line of work introduces additional parameters to acquire new knowledge (Wang et al., 2022b;c;a; Smith et al., 2023). Determining which set of newly added parameters to use during inference remains challenging. Additionally, the performance of such works is highly dependent on the capacity and flexibility of the added parameters, where some works only get a marginal improvement over the pretrained model (Janson et al., 2022). Our work focuses on modifying the pre-trained models themselves, and shares some similarities with weight regularization methods Kirkpatrick et al. (2017); Aljundi et al. (2018) where an importance or relevance score is estimated for the model's parameters. A clear distinction is that the parameter importance score is estimated *after* learning a given task and used to *prevent* changing those important parameters. Differently, our approach estimates the parameter's relevance score for a new task *before* starting the learning process. Our selection is to identify which parameters to *update*. Finally, the majority of these approaches focus on defying forgetting in the learned sequence, with no consideration for the forgetting of pre-trained knowledge. Further, they do not scale to preserving pretrained knowledge, as they either require access to the pretraining dataset (Aljundi et al., 2018; Kirkpatrick et al., 2017; Chaudhry et al., 2019b) or a duplicate storage of the pretrained model (Li & Hoiem, 2017; Asadi et al., 2023). In contrast, we consider the accumulation of knowledge, including the pre-trained and newly acquired knowledge, without any task identification and extra storage of model weights.

**Finetuning with Knowledge Preservation.** It is usually observed that when finetuning foundation models on new tasks, the generic knowledge and transferability are severely deteriorated. Recently, some works (Meng et al., 2022; Ilharco et al., 2022; Xiang et al., 2023; Khattak et al., 2023; Zheng et al., 2023) started to tackle the issue of updating large pretrained models while preserving their transferability and the generalizability. Among them, Meng et al. (2022); Ilharco et al. (2022) proposes model editing algorithms, where the models are first analyzed to pick specific layers to edit, and then algebra-based or meta-learning based methods are applied to the weight of the local layer. Usually, a local set is utilized to preserve the background knowledge. Additionally, Xiang et al. (2023) proposed to finetune language models for embodied tasks while maintaining their generalization ability to handle unseen embodied tasks. They suggested fine-tuning language models with LoRa (Hu et al., 2022), i.e., low rank updates, to ensure compute efficiency, while applying EWC regularization Kirkpatrick et al. (2017) to reduce forgetting of the pretrained knowledge. On the multimodal models end, Zheng et al. (2023) considered to prevent zero-shot transfer degradation in the continual learning of CLIP by performing distillation on the pre-trained model weights. However, it requires access to a massive dataset to represent the pre-training distribution, which is not a trivial assumption and far from being computationally efficient. In this work, we aim to update foundation models, such as CLIP, continually to recognize additional concepts and preserve their transferability, while striving for efficiency.

## 3  CONTINUAL LEARNING FROM PRE-TRAINED MODELS

In Class Incremental Learning (CIL), we are given a dataset $D_{\text{train}}^t = \{x_k, y_k\}_{k=1}^{N_t} \sim \mathcal{D}^t$ sampled from a task-specific distribution $\mathcal{D}^t$ for each task $t \in \{1, \ldots, T\}$ sequentially, where $X_{\text{train}}^t = \{x_k\}_{k=1}^{N_t}$ is a set of images and $Y_{\text{train}}^t = \{y_k\}_{k=1}^{N_t}$ is the set of the corresponding labels with $y_k \in Y_{\text{train}}^t$. Here $Y_{\text{train}}^t$ is the label space of task $t$. Note that while we focus on image-based data, our method can be extended to any modality. We are given a model parameterized by $\theta$ pre-trained on a vast pre-training dataset $D_p \sim \mathcal{D}_p$ sampled from the pre-training distribution, which is inaccessible during the CIL procedure. During the learning of each task, the model parameters $\theta$ are to be optimized to minimize a loss function $\mathcal{L}$ on the current training set $D_{\text{train}}^t$. The loss function depends on the task at hand and the model deployed. For CLIP model (Radford et al., 2021) and image text pairs data, we deploy

the same contrastive loss used for CLIP pretraining. After the learning of each task, we evaluate our model on both the validation set of the seen distributions of the CIL sequence $D_{\text{test}}^{1:t}$, where $D_{\text{test}}^{t} \sim \mathcal{D}^{t}$, and a small control set $D_{\text{control}} \sim \mathcal{D}_{p}$ sampled from the pre-training distribution.

## 4 METHODOLOGY: SELECTIVE PARAMETER UPDATE

Most existing continual learning methods that start from randomly initialized models optimize all parameters equally, as such a starting point has no knowledge or relevance to the task being learned. However, foundation models often have a reasonable initial performance on novel tasks, indicating some pre-existing knowledge relevant to these tasks. With the strive for efficiency and the preservation of the generic knowledge, we suggest identifying a small set of parameters corresponding to these pre-existing knowledge and only updating them instead of modifying all the pre-trained model parameters. We now introduce how to localize the update to specific layers and how to identify a sparse set of specialized parameters to be optimized.

**Localization.** The objective of our work is to accumulate new knowledge without catastrophically forgetting the generic knowledge. To achieve this, we introduce a method that performs local changes restricted to specific layers in the pre-trained transformer backbones. As shown in Figure 1, a transformer block contains a multi-head attention block and a two-layer MLP block. Recent research on transformer analysis (Geva et al., 2021) has shown that MLP blocks emulate key-value neural memories, where the first layer of MLP acts as memory keys, operating as pattern detectors. Each individual key corresponds to a specific pattern seen in the input data. Whereas, the second layer learns the distribution over the detected patterns. Our work aims to add, update, or refine current knowledge embedded in the model, and with the

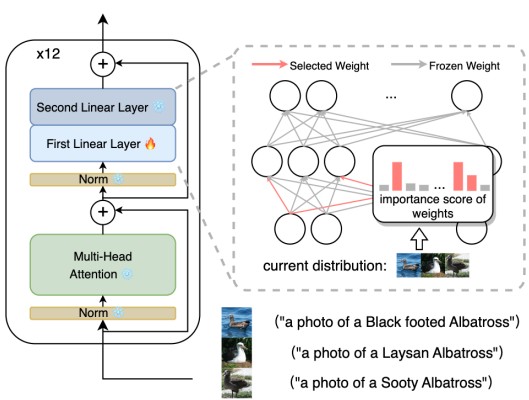

Figure 1: We first localize our update to the first layer of MLP blocks, and then select a sparse set of parameters specialized to the new task to update.

analogy to the key-value memories, we opt for refining the keys (corresponding to pattern detectors) to accommodate the new information. Empirically, we investigated whether we need to change the patterns' distributions represented by the second layer as well, and it turned out that updating the first layer is sufficient and more effective, as we shall show in the experiments Section 5.5. With this in mind, we localize the model updates to the first layer of the MLP in each transformer block. With such localization, our candidate parameters to change are only around one third of the total parameters.

**Parameter Selection.** Pre-trained foundational models have inherent knowledge, as evidenced by their capacity to execute diverse tasks without fine-tuning. Moreover, recent investigations (Geva et al., 2021; 2022; Bills et al., 2023) have unveiled the correlation between the concepts and specific neurons' output in foundation language models. Therefore, we hypothesize that there exists some sort of modulation and specialization among specific neurons and their corresponding parameters in foundation models. Upon these, we propose to identify which parameters in the first MLP layer are specialized on the task at hand based on a scoring function. As shown in Figure 1, we select parameters associated with top scores and minimize the new task loss by *only* updating those selected parameters. In practice, we only make small changes to 3% of the total parameters, leading to efficiency and effectiveness in preventing forgetting, as we shall show in later experiments. We now formally outline our parameter selection methodology.

We receive the current task dataset $D^{t}$ representing a task $t$ in a continual learning sequence and localize the updates to the first MLP layer $\theta^{l}$ for each transformer block, where $l$ denotes the localized first layer indexed over transformer blocks. We aim to define an element-wise scoring function $\mathcal{S}(\theta_{i,j}^{l}, D^{t})$, for each parameter in a localized layer $\theta_{i,j}^{l}$; $i, j$ refers to the parameter connecting an input element $i$ (the $i$-th output entry of the attention layer) to the neuron $j$ in the first MLP layer. We propose to select a subset of parameters $\theta_{U}^{l} \subseteq \theta^{l}$ that has the largest scores $\{\mathcal{S}(\theta_{i,j}^{l}, D^{t})\}$, subject to $\frac{|\theta_{U}^{l}|}{|\theta^{l}|} = r$, where $|\cdot|$ is the parameter size and $r$ is the selection rate. This set is then expected

to combine the most relevant parameters to the current task represented by the dataset $D^t$. We select parameters regardless of their corresponding neurons and ablate the effect of selecting the entire parameters of identified neurons in Appendix B. In the sequel, we present two variants of our parameter scoring function. For clarity, the presentation of the method is focused on $\theta^l$, and it can be generalized to a plural of selected layers covering all transformer blocks.

**Gradient-Based Scoring Function.** We aim to identify which parameters are more relevant to the new task at hand. We formulate this as finding parameters where small changes to their values could lead to a greater improvement in the task performance. When achieving this, we only make small changes to the model and thus can preserve the generic knowledge while improving the new task performance. As a proxy to the new task performance, we use the task loss function and approximate the changes in this loss function amid small changes in each parameter. Specifically, we can approximate the change in the loss function upon small changes $\delta$ in the parameters' values with

$$\mathcal{L}(\theta^l + \delta; x_k) - \mathcal{L}(\theta^l; x_k) \approx \sum_{i,j} g_{ij}(x_k)\delta_{ij}, \tag{1}$$

where $\mathcal{L}$ is the loss function, $g_{ij}(x_k) = \frac{\partial(\mathcal{L}(\theta^l; x_k))}{\partial \theta^l_{ij}}$ is the gradient of the loss function regarding the parameter $\theta^l_{ij}$ evaluated at the data point $x_k \in D^t$. The above first-order approximation suggests that small changes made to parameters with the largest gradient magnitude $\| g_{ij} \|$ in the opposite direction of the gradient would incur a larger reduction in the loss function, and hence greater improvements with minor changes.

Following this, we define our scoring function as:

$$\mathcal{S}(\theta^l_{ij}, D^t) = \| \frac{1}{N'_t} \sum_{k=1}^{N'_t} g_{ij}(x_k) \|, \tag{2}$$

where $N'_t$ is the number of samples we use to compute the gradient. Note that $N'_t$ can be much smaller than the total number of samples in the dataset, $N_t$ as we shall show in the experiments 5.3.

**Learnable Scoring Function.** Although the gradient is an efficient approximation of the parameters' relevance to the task at hand, selecting parameters independently based on their gradient magnitude might not consider the contribution of the parameters together when updated, and can potentially cause redundancy in the selection. To explore this, we propose to involve an optional optimization stage to adjust the scoring function based on the initial gradient values. Specifically, for parameters $\theta^l \in \mathbb{R}^{m \times n}$, we define $\boldsymbol{S} \in \mathbb{R}^{m \times n}$ to be the learnable parameters scores. We initialize $\boldsymbol{S}$ with the gradients computed on the current task, where $\boldsymbol{S}_{ij} = \frac{1}{N'_t} \sum_{k=1}^{N'_t} g_{ij}(x_k)$. We consider the estimated gradient as the bases for a target update of the model parameters and construct an imaginary update:

$$\theta^{l\prime} = \theta^l - \mu \cdot \boldsymbol{S}, \tag{3}$$

where $\mu$ is the update step size (learning rate). We then optimize $\boldsymbol{S}$ by minimizing the task loss $\mathcal{L}$ and an additional $L_1$ loss ($\|\boldsymbol{S}\|_1$)

$$\boldsymbol{S}' = \arg\min_{\boldsymbol{S}} \mathcal{L}(\theta^{l\prime}; D^t) + \lambda\|\boldsymbol{S}\|_1, \tag{4}$$

where $\lambda$ is a hyperparameter that weighs the contribution of $L_1$ loss. $L_1$ loss is introduced to encourage sparsity in the estimated scores, guiding the optimization to tolerate parameters with large gradient magnitude (and hence large initial scores) when proven relevant to the minimization of the task loss while zeroing out gradients of irrelevant or redundant parameters.

We optimize $\boldsymbol{S}$ for a few epochs. Then, we define $\mathcal{S}(\theta^l_{ij}, D^t) = \boldsymbol{S}'_{i,j}$ and select top $r$ parameters as the most relevant parameters for the task at hand. Note that here we estimate parameters scores for one selected layer $\theta^l$, but the formulation can generalize to an arbitrary number of layers.

While this holds the promise of a better selection, it requires more computation due to the additional optimization phase of $\boldsymbol{S}$ compared to the gradient scores. For efficiency consideration, we report our main results in Table 5.2 with the scoring function of gradient approximation, and present the efficacy of this optional stage in Section 5.4.

**Sparse update.** Upon selecting the relevant parameters $\theta_U = \{\theta^l_U\}$, we freeze all other model parameters and learn the current dataset $D^t$ by only optimizing $\theta_U$.

Following the current practice in class incremental learning methods, (Asadi et al., 2023; Ebrahimi et al., 2020; Wang et al., 2022c) we deploy a replay buffer to reduce the forgetting across the new tasks sequence. We keep a replay buffer $\mathcal{M}$ of a fixed size, and sample batches from it of the same

size as the batch from the current dataset at each optimization step. We update the replay buffer at the end of learning of each task by reservoir sampling (Chaudhry et al., 2019b).

Our final objective function at task $t$ can be written as

$$\min_{\theta_U} \mathcal{L}(\theta; D_{\text{train}}^t) + \mathcal{L}(\theta; \mathcal{M}) \tag{5}$$

where $\mathcal{L}(\theta; D)$ is the current task loss computed on the training set $D^t$.

**Algorithm applicability.** Our algorithm involves three key steps: localizing update layers, selecting relevant parameters, and training on the new task with sparse updates. It is important to note that while we primarily delve into the localization within the transformer architecture, the concept of selectively updating certain layers while keeping others frozen to achieve efficiency and comparable performance is not confined to this architecture alone.(Santurkar et al., 2021; Caccia et al., 2020). Should the need arises to extend our approach to different architectures, the first step of our methodology can be readily adapted. Furthermore, the processes of parameter selection and sparse updates remain architecture-agnostic, making them versatile across various model structures. We refer to our method as SPU short for Selective Parameter Finetuning.

## 5 EXPERIMENTS

We evaluate our proposed framework on various datasets compared to different methods and baselines in Section 5.2, and analyze different components of our method and ablate our design choices in Section 5.5. We provide further ablations on defying generic knowledge loss in the Appendix.

### 5.1 SETUP

**Backbone.** We assess the efficacy of our approach through its application to vision-language classification tasks, given the straightforward measurement of knowledge base in such tasks. we choose the pre-trained CLIP-ViT/B-16 (Radford et al., 2021) as our backbone.

**Datasets.** We assess the performance of our algorithms on a total of six datasets— four fine-grained datasets (Birdsnap (Berg et al., 2014), CUB-200-2011 (Wah et al., 2011), FGVC-Aircraft (Maji et al., 2013), Stanford Cars (Krause et al., 2013)), one coarse dataset (CIFAR100 Krizhevsky et al. (2009)), and one out of distribution dataset (GTSRB (Stallkamp et al., 2012)) . These datasets were chosen primarily based on their initially low zero-shot performance with CLIP pre-trained models. To form the continual learning sequences, we split each dataset into 10 subsets with disjoint classes composing 10 tasks. For methods that leverage a replay buffer, we use a buffer size of around 4% of the dataset size. Ablation study of buffer size is shown in Section 5.5. For more comprehensive information, including detailed statistics, data sources, and implementation details, please refer to Appendix A.

**Baselines.** We conduct a comprehensive comparison of our method against various baselines to demonstrate its effectiveness. Firstly, we evaluate our approach against the fine-tuning baseline of CLIP, FLYP (Goyal et al., 2023). We further integrate classical continual learning components to evaluate their performance on the CLIP backbone, including reservoir replay buffer (Chaudhry et al., 2019b), weight regularization method, MAS (Aljundi et al., 2018), and functional regularization methods LwF (Li & Hoiem, 2017) and PRD (Asadi et al., 2023). We combine these functional regularization methods with experience replay. Furthermore, against recent work that also prioritize the preservation of generic knowledge, namely ZSCL (Zheng et al., 2023). Finally, we compare our method with the latest pre-trained model based continual learning techniques, namely L2P (Wang et al., 2022c) and DualPrompt (Wang et al., 2022b). We find that these two methods struggle with CLIP backbone and fail to produce sensible results despite our best attempts to tune their hyperparameters. Results, evaluation with ImageNet pretrained backbones of these methods, and discussion are in Appendix A.

**Evaluation.** We measure the Average Accuracy (Avg. Acc.) at the end of the class-incremental process, as well as the forgetting rate following prior arts (Chaudhry et al., 2019b;a). Additionally, we aim to understand how the knowledge base shifts as we continually update the pre-trained models. To achieve this, and similar to (Ilharco et al., 2022), we evaluate the model on a diverse dataset with generic knowledge before and after the continual learning procedure. Specifically, we report the zero-shot classification accuracy on the validation set of ImageNet (Deng et al., 2009) as the control set Accuracy (C.), and compare it with that from the frozen pre-trained models.

Table 1: Average Accuracy (Acc.), Forgetting (F.), and control set Accuracy (C.) of our method SPU and baselines on 6 CIL sequences, demonstrating our superior knowledge accumulation and preservation. We highlight parameter efficiency via parameters size and learnable parameters rate, and data efficiency via data use.

| | | Frozen | FLYP | + MAS | + ER | + ER + LwF | + ER + PRD | ZSCL | SPU (Ours) |
|---|---|---|---|---|---|---|---|---|---|
| Parameter Size | | 149.5M | 149.5M | 149.5M | 149.5M | 299M | 299M | 299M | 149.5M |
| Learnable Parameter Rate | | 0.00 | 1.00 | 1.00 | 1.00 | 0.5 | 0.5 | 0.5 | 0.03 |
| Data Source | | - | current task | current task | current task, buffer | current task, buffer | current task, buffer | current task, CC12M | current task, buffer |
| Aircraft | Acc. | 24.45 | 18.63 | 33.69 | 41.42 | 36.08 | 37.11 | 30.96 | **44.43** |
| | F. | - | 39.93 | 27.50 | 31.48 | 18.12 | 17.35 | 15.65 | 14.42 |
| | C. | 63.55 | 41.04 | 61.09 | 50.41 | 63.06 | 63.38 | **65.53** | 63.48 |
| Bird snap | Acc. | 43.20 | 44.06 | 47.42 | **56.22** | 50.23 | 51.34 | 49.85 | 55.35 |
| | F. | - | 23.43 | 17.12 | 21.63 | 10.20 | 9.45 | 13.28 | 12.78 |
| | C. | 63.55 | 51.06 | 60.05 | 56.72 | 62.08 | 62.85 | **63.13** | 61.94 |
| Cars | Acc. | 64.63 | 51.64 | 69.43 | 69.08 | 72.56 | 74.08 | 67.79 | **77.51** |
| | F. | - | 25.65 | 9.18 | 16.42 | 4.04 | 3.75 | 8.27 | **3.26** |
| | C. | 63.55 | 52.25 | 61.17 | 58.07 | 62.59 | 62.96 | 62.90 | **63.42** |
| CIFAR 100 | Acc. | 68.25 | 46.26 | 63.88 | 82.86 | 74.32 | 79.66 | 80.50 | **83.99** |
| | F. | - | 37.78 | 21.16 | 3.41 | 8.16 | 3.10 | 1.05 | **-0.39** |
| | C. | 63.55 | 26.53 | 49.35 | 42.10 | 55.71 | 59.01 | **61.90** | 61.38 |
| CUB | Acc. | 55.13 | 45.74 | 61.72 | 64.07 | 65.11 | 65.92 | 61.09 | **71.51** |
| | F. | - | 26.62 | 12.05 | 17.72 | 5.90 | 6.55 | 7.69 | **4.84** |
| | C. | 63.55 | 44.30 | 57.35 | 51.30 | 62.05 | 62.09 | 62.78 | **62.87** |
| GTSRB | Acc. | 43.38 | 21.76 | 42.04 | **96.28** | 53.56 | 63.00 | 62.92 | 94.25 |
| | F. | - | 55.48 | 25.38 | -7.40 | 11.86 | 12.44 | 13.54 | **-7.87** |
| | C. | 63.55 | 1.59 | 42.06 | 17.34 | 57.99 | 61.04 | **62.92** | 62.55 |
| Avg. Acc. Impr. (↑) | | 0.0 | -11.82 | 3.19 | 18.48 | 8.80 | 12.01 | 9.01 | **21.34** |
| Avg. F. (↓) | | - | 34.81 | 18.73 | 13.88 | 9.71 | 8.77 | 9.91 | **4.51** |
| Avg. C. Drop (↓) | | 0.0 | 27.42 | 8.37 | 17.56 | 2.97 | 1.66 | **0.36** | 0.94 |

To provide a comprehensive view of baseline performance across all 6 datasets, we present the improvement of Average Accuracy (Avg. Acc. Impr.) from the frozen pre-trained model across these datasets, as well as the average Forgetting rate (Avg. F.) and the average loss of control set accuracy (Avg. C. Drop.).

**Implementation Details.** We follow (Goyal et al., 2023) to both perform selection and sparse update on the visual tower and text tower of the CLIP model, and use contrastive loss as our loss function. Within our algorithm, we use a selection rate of 10%, which optimally balances learning and forgetting. We perform an ablation study on the selection rate in section 5.5. More implementation details of all baselines can be found in Appendix A.

## 5.2 RESULTS

We present the comparison between our method with baselines in Table 1. In the subsequent sections, we delve into our observations from the dual lenses of learning and forgetting.

**Learning.** Regarding the accumulation of novel information, we achieve state-of-the-art results in four out of six datasets, *i.e.*, Aircraft, Cars, CIFAR100, and CUB, and comparable results in Birdsnap and GTSRB, with a notable 1% - 10% average improvement over the existing continual learning methods. Among the continual learning baselines we've compared to, FLYP+ER stands as the only comparable contender in terms of average accuracy. However, it exhibits a significant drawback in the form of forgetting, averaging at 13.88% in the forgetting of the current dataset, and a notable decrease of 17.56% in average control set accuracy. In stark contrast, our approach demonstrates a much more favorable performance regarding these two key metrics, with a mere 4.90% forgetting rate and 0.94% decrease in average control set accuracy.

**Forgetting.** While accumulating novel knowledge remains paramount, we also prioritize the knowledge retention. This includes both the fading of newly accumulated information within the task

Table 2: Our SPU efficiency compared to full finetuning. 1) CIL performance with one pass to the data (1 epoch), 2) parameters update time, 3) forward and backward passes time.

| | Aircraft | | | Birdsnap | | | Cars | | | CUB | | | Optimizer Step Time (ms) | Batch Time (s) |
|---|---|---|---|---|---|---|---|---|---|---|---|---|---|---|
| | Acc. | F. | C. | Acc. | F. | C. | Acc. | F. | C. | Acc. | F. | C. | | |
| FLYP+ER | **31.32** | **10.04** | 61.17 | **55.08** | 11.87 | 60.62 | 65.30 | 4.66 | 61.47 | 58.27 | 6.06 | 60.02 | 83.544 | 0.363 |
| SPU (Ours) | 31.17 | 12.20 | **63.67** | 52.21 | **7.15** | **64.12** | **69.98** | **2.72** | 63.52 | **61.25** | **5.68** | **63.51** | **4.167** | **0.155** |

sequence, measured by the F. metric, and the retention of pre-training phase knowledge, measured by the C. metric. Our method yields a marked improvement across all datasets, resulting in a substantial margin in both metrics. Notably, distillation-based methods such as FLYP+ER+LwF/PRD and ZSCL generally perform good at preserving the pre-trained knowledge, all displaying control set accuracy drop of less than 3%. However, their flexibility in learning the new tasks, as indicated by their average accuracy, remains limited, reflecting a discernible gap of over 8% when compared to our method.

**Fine-grained Datasets.** The diverse characteristics of various datasets also lead to distinct behaviors. Across fine-grained datasets like Aircraft, Cars, and CUB, we achieve SOTA average accuracy, outperforming the baselines by around 3%, while demonstrating minimal degradation in control set accuracy of less than 1%.

**Out of Distribution Dataset.** We consider GSTRB as out of distribution regarding CLIP pretraining, as it is the only considered dataset with CLIP zero shot performance significantly lower than the performance of a linear classifier trained on ResNet50 features Radford et al. (2021). In our results, GSTRB proves an outlier for SOTA CIL methods with significantly low Acc., our method proves robust. FLYP+ER achieves an average accuracy of 96.28% in GTSRB, but at the expense of a 46.21% control set accuracy, equating to around 60% loss in the control set accuracy, referring to a large decay in the generic knowledge after learning such out of distribution dataset. In contrast, our proposed method achieves competitive accuracy, concurrently delivering small control set loss of around 1%, signifying minimal loss of generic knowledge.

**Coarse Dataset.** In contrast, in the case of the coarser CIFAR100 dataset, we still achieve an impressive SOTA learning accuracy of 83.99%, albeit with a marginal trade-off of approximately 2% in control set accuracy. Even with this reduction, our approach stands out as significant compared to other continual learning techniques that experience losses of generic knowledge ranging from 4% to 21%. This phenomenon can be attributed to that CIFAR100 encapsulates a degree of generic knowledge, possibly causing inference in the information on control sets like ImageNet1k.

## 5.3 EFFICIENCY

We introduce the total parameter size and the rate of learnable parameters in Table 1. While most of the current methods necessitate a complete parameter update, our approach only requires an update of a sparse subset of parameters, which only consists of 3% of the total model's parameters. This characteristic contributes to both computational and data efficiency. We perform experiments that only train one epoch with our algorithm and FLYP + ER on four fine-grained datasets, and measure the average per optimization step time and per batch time on the same machine by `line profiler` (Crall & Kern, 2023). The Results are shown in Table 2. With only one epoch of the experiment, we achieve better results on cleaner fine-grained datasets like CUB and Cats. We achieve 69.98% accuracy in Cars and 61.25% accuracy in CUB, which is only around 8% lower than the main results in Table 1. And it is 4% higher than the baseline of FLYP+ER. From the perspective of running time, we reduce the per optimization step time from 85.433 ms to 4.167 ms. Although we need an extra masking operation, we reduced the average batch time from 0.363s to 0.155s. Note that the masking step can be done prior to the training, further reducing the batch processing time.

We further perform an ablation study on the number of samples $N_t'$ used to approximate the scoring function. Results show that our method can still have good performance even when using only one batch of samples for the approximation. This implies that the computation of the scoring function is also efficient which does not require a full pass of the data prior to the training, and can be done transparently with the first received batch. Details are shown in Appendix B.

## 5.4 SCORING FUNCTION

We presented two scoring functions, one is gradient-based, and the other is learnable in Section 4. Here we study their effect on the different metrics, shown in Table 3. While the average accuracy (Acc.) is similar among the two scoring mechanisms, the learnable one tends to have slightly less

Table 3: Our method SPU performance when updated parameters are selected based on the gradient-based scoring function compared to learnable scoring function.

| Scoring Function | Aircraft | | | Birdsnap | | | Cars | | | CIFAR100 | | | CUB | | | GTSRB | | |
|---|---|---|---|---|---|---|---|---|---|---|---|---|---|---|---|---|---|---|
| | Acc. | F. | C. | Acc. | F. | C. | Acc. | F. | C. | Acc. | F. | C. | Acc. | F. | C. | Acc. | F. | C. |
| Gradient-based | **44.43** | **14.42** | 63.48 | **55.35** | **12.78** | 61.94 | **77.51** | **3.26** | 63.42 | 83.99 | **-0.39** | 61.38 | **71.51** | 4.84 | 62.87 | **94.25** | **-7.87** | 62.55 |
| Learnable score | 43.95 | 14.80 | **63.58** | 54.23 | 13.10 | **62.23** | 76.92 | 3.56 | **63.43** | **84.30** | -1.07 | **62.08** | 71.11 | **4.78** | **62.98** | 92.41 | -7.33 | **62.63** |

forgetting and less control set accuracy loss. This suggests that the learnable variant can alleviate some redundancy in parameters scoring and hence suffers fewer changes on the pretrained parameters. Overall, the simple and efficient gradient-based scoring is quite robust compared to the learnable one. Implementation details can be found in Appendix A.

## 5.5 ABLATION STUDY

In this section, we perform ablation studies on the individual components comprising our algorithm. These analyses serve to validate the rationale behind our design of these components. We only show the average results here. Refer to Appendix B for more details

**Parameters localization.** We compare localizing the update to the of the first MLP layer parameters (our choice) to that of the second layer and to localizing both layers together in Table 4. All variants update the same number of parameters. Updating parameters from the second layer suffers double the generic knowledge loss compared to that of the First layer parameters.

Table 4: Ablation on localized layers

| Localized Layer | Avg. Acc. Impr. | Avg. F. | Avg. C. Drop |
|---|---|---|---|
| First | **21.34** | 4.51 | **0.94** |
| Second | 20.51 | **4.35** | 1.85 |
| Both | 21.18 | 4.88 | 1.51 |

Updating parameters in both layers is also worse in both forgetting and control set accuracy than localizing the updates to the first layer only. We conclude that localizing the updates to selected parameters of the first layer only is sufficient to achieve the best trade-offs.

**Selection rate.** Table 15 illustrates the variants of our method under varying selection rates applied to the first layer of MLP blocks. Across all selection rates, our method demonstrates competitive average accuracy, forgetting, and control set accuracy when compared with other baselines in Table 1. Even with a 0.5 selection rate, the learnable parameters comprise

Table 5: Selection rate ablation

| Selection Rate $r$ | Avg. Acc. Impr. | Avg. F. | Avg. C. Drop |
|---|---|---|---|
| 0.01 | 17.70 | **3.10** | 1.11 |
| 0.10 | 21.34 | 4.51 | **0.94** |
| 0.50 | **21.73** | 7.76 | 0.95 |

only 30% of the total parameters. We note that as the selection rate increases, there is a marginal enhancement in learning performance, but accompanied by a compromise in forgetting. For instance, raising from 0.1 to 0.5 selection rate, the Average Accuracy improves around 0.5% and the forgetting also raises around 3%. Therefore, we opt for a selection rate of 0.1, which gives the best trade-off between the accumulation of the new knowledge and preservation of the pre-trained knowledge.

**Buffer size.** In Table 1, we present the outcomes of our approach using a buffer size equivalent to 4% of the total dataset size. Table 6 shows our performance over an array of buffer sizes, ranging from 1% to 4% of the total dataset size, compared with ER. Evidently, our algorithm excels in preserving pre-training knowledge across all buffer sizes, all with less than 1% drop in control set accuracy. As we decrease the buffer size, FLYP+ER encounters

Table 6: Buffer size ablation

| Method | Buffer Size | Avg. Acc. Impr. | Avg. F. | Avg. C. Drop |
|---|---|---|---|---|
| FLYP+ER | 1% | 8.97 | 22.27 | 19.18 |
| | 2% | 13.24 | 19.35 | 18.24 |
| | 4% | **18.48** | **13.88** | **17.56** |
| SPU | 1% | 16.18 | 10.28 | 1.00 |
| | 2% | 18.63 | 8.14 | 0.96 |
| | 4% | **21.34** | **4.51** | **0.94** |

substantial influence; our method with 1% buffer size doubles Avg. Acc. improvement of FLYP+ER with 1% buffer and suffers 50% less forgetting while only tolerating 1% control set accuracy loss.

## 6 DISCUSSION

With the rise of advanced foundation models pretrained on vast datasets, we propose a method that preserves pre-learned information in continual learning. We base on the fact that foundation models already have initial knowledge for the task in hand, and identify specific model layers and parameters corresponding to this knowledge for sparse updates. As such, we perform small update for the model to cope with the new knowledge while preserving the previously acquired generic knowledge. We evaluate our method extensively and show superior performance. However, our current method operates unidirectional, and future research could explore knowledge accumulation across diverse domains. Additionally, expanding our focus from discriminative to generative tasks would enhance the applicability of our techniques.

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

## A    IMPLEMENTATION DETAILS

### A.1    DATASET

*Birdsnap* Birdsnap is a large bird dataset originally consisting of 49,829 images from 500 bird species with 47,386 images used for training and 2,443 images used for testing. We download the dataset from the official link. We follow the official train-test split. We use a fixed buffer of size 1,500 for this dataset.

*CUB-200-2011* The Caltech-UCSD Birds-200-2011 (CUB-200-2011) dataset is for fine-grained visual categorization task. It contains 11,788 images of 200 subcategories belonging to birds, 5,994 for training and 5,794 for testing. We use the Hugging Face implementation of the dataloader. We use a fixed buffer of size 240 for this dataset.

*CIFAR100* This dataset has 100 classes containing 600 images each. There are 500 training images and 100 testing images per class. We use the PyTorch implementation of the dataloader. We used a fixed buffer of 2,000 for this dataset.

*FGVCAircraft* The dataset contains 10,200 images of aircraft, with 100 images for each of 102 different aircraft model variants, most of which are airplanes. The data is divided into three equally-sized training, validation and test subsets. We use the PyTorch implementation of the dataloader, where train and valid set are used for training, and the test set is used for testing. We use a fixed buffer of size 250 for this dataset.

*Stanford Cars* The Cars dataset contains 16,185 images of 196 classes of cars. The data is split into 8,144 training images and 8,041 testing images, where each class has been split roughly in a 50-50 split. Classes are typically at the level of Make, Model, Year, ex. 2012 Tesla Model S or 2012 BMW M3 coupe. We use the Hugging Face implementation of the dataloader. We use a fixed buffer of size 240 for this dataset.

*GTSRB* This dataset is designed for recognition of traffic signs. By the time we download it, it contains 43 classes with 26,640 training samples and 12,630 testing samples. We use the PyTorch implementation of the dataloader. We used a fixed buffer of 1,000 for this dataset.

For each dataset, during the training, we use the prompt `a photo of {}` with class name as text inputs. We evaluate each baseline on the test set using the original prompts and ensembling strategy provided by  Radford et al. (2021).

### A.2    BASELINE DETAILS AND HYPER-PARAMETERS

For our algorithm, we use PyTorch implemented AdamW optimizer  (Loshchilov & Hutter, 2017) and learning rate scheduler of Cosine Annealing with Warmup  (Loshchilov & Hutter, 2016) for our algorithm, as well as FLYP combined with ER and other CL regularization methods. We use a learning rate of 7.5e-6 and train for 10 epochs for all datasets. We report results based on an average of 5 different random seeds. We run all our experiments on one single Nvidia A100 GPU.

Here's the implementation for other baselines.

**FLYP** For all FLYP based baselines, we tune the learning rate in [2.5e-6, 5e-6, 7.5e-6] and training epochs in [5,10,15] and report the best results

**FLYP + MAS** We follow avalanche  (Lomonaco et al., 2021) to implement MAS regularier with FLYP. To normalize the magnitude of MAS importance weights so that it copes with the large-scale architecture, we normalize the estimated importance wights by its maximum value. We tune the scaling factor of MAS loss in [0.01, 0.05, 0.1] and report the best.

**FLYP + ER + LwF/PRD**  For the distillation-based baselines, we follow the implementation of avalanche and official implementation of PRD. We further tune temperature in [0.01,0.1,1.0,5.0] and loss scaling factor in [0.01, 0.05, 0.1] and report the best results.

**L2P, Dualprompt** For the prompt-based baselines, we deploy them with the CLIP pre-trained ViT provided by timm library. We believe that these methods are highly tailored for ImageNet pretrained transformers and do not scale to other backbones, leading to surprisingly bad performance when combined with CLIP in sight of our best efforts to tune the hyper-parameters carefully. The results are

Table 7: L2P and DualPrompt performance with CLIP-ViT backbone

| method | Aircraft | | Birdsnap | | Cars | | CIFAR100 | | CUB | | GTSRB | |
|---|---|---|---|---|---|---|---|---|---|---|---|---|
| | Acc. | F. | Acc. | F. | Acc. | F. | Acc. | F. | Acc. | F. | Acc. | F. |
| L2P | 3.06 | 15.80 | 2.53 | 13.31 | 1.41 | 13.34 | 7.28 | 37.74 | 5.28 | 17.51 | 7.19 | 39.40 |
| DualPrompt | 3.27 | 10.27 | 2.91 | 3.49 | 1.50 | 7.14 | 3.76 | 28.63 | 3.64 | 11.64 | 6.63 | 46.05 |

Table 8: L2P and DualPrompt performance accuracy on CIRAR100 with different pre-trained weights.

| ViT Pre-trained Weights Source | Pretrained on ImageNet 21k | Released by CLIP finetuned on ImageNet 1k | Released by CLIP finetuned on ImageNet 12k | Released by CLIP |
|---|---|---|---|---|
| L2P | 83.32 | 12.82 | 11.73 | 7.28 |
| DualPrompt | 82.55 | 52.72 | 25.29 | 3.76 |

shown in table 7. To validate this, we report the continual learning average accuracy of the sequence 10 split CIFAR100 with different pre-trained weights in Table 8.

## A.3 IMPLEMENTATION DETAILS OF THE LEARNABLE SCORING FUNCTION

In practice, we optimize the learnable score matrix $S$ for 5 epochs, with the learning rate of $5e-4$ and step size $\mu = 5e-4$. We set the $L_1$ loss coefficient $\lambda = 1e-3$

## B FULL RESULTS OF ABLATION STUDY

**Parameter Localization.** Table 9 shows the full results of variants of our method in editing different layers. In the baseline named "First" or "Second", we free all other layers except first or second layers of each of the MLP blocks respectively, and select 10% elements to update. In the baseline named "Both", we unfreeze both layers of the MLP block, and select 5% elements on each layer to update to equate the same number of parameters as in the single layer variant.

Table 9: Full results of ablation on parameter localization

| | Aircraft | | | Birdsnap | | | Cars | | | CIFAR100 | | | CUB | | | GTSRB | | |
|---|---|---|---|---|---|---|---|---|---|---|---|---|---|---|---|---|---|---|
| | Acc. | F. | C. | Acc. | F. | C. | Acc. | F. | C. | Acc. | F. | C. | Acc. | F. | C. | Acc. | F. | C. |
| First | 44.43 | 14.42 | 63.48 | 55.35 | 12.78 | 61.94 | 77.51 | 3.26 | 63.42 | 83.99 | -0.39 | 61.38 | 71.51 | 4.84 | 62.87 | 94.25 | -7.87 | 62.55 |
| Second | 43.32 | 14.02 | 63.24 | 54.98 | 12.25 | 61.17 | 76.91 | 3.24 | 62.77 | 83.59 | -0.42 | 59.57 | 70.00 | 5.40 | 62.19 | 93.32 | -8.38 | 61.24 |
| Both | 44.21 | 14.78 | 63.32 | 55.10 | 13.46 | 61.32 | 77.25 | 3.79 | 63.13 | 84.15 | -0.31 | 60.47 | 71.23 | 5.42 | 62.34 | 94.18 | -7.85 | 61.65 |

**Selection Strategy.** Table 11 shows the full results of variants of our methods in selection strategy. In the baseline named "Weight" we compute an element-wise scoring function by Equation 2, and select the top 10% entries of each weight matrix to update. In the baseline named "Neuron", we compute a row-wise scoring function based on the row summation of the element-wise scoring function by Equation 2. Then we select the 10% rows of each weight matrix to update.

Aljundi et al. (2019) put forth a technique to calculate row-wise importance scores to perform neuron-based selection.

We conduct variants of our the selection strategy and present the results in Table 10. In neuron-based selection, we compute the importance score as the summation of the importance score of each row. We find that weight-based selection yields slightly improved learning performance while exhibiting a marginal decrease in hold-out accuracy. Nonetheless, the overall performance trends remain comparable between the two strategies. This observation highlights the robustness of our localization and importance scoring methods to any of the selection strategy.

Table 10: Ablation on selection strategy.

| Update Layer | Avg. Acc. Impr. | Avg. F. | Avg. C. Drop |
|---|---|---|---|
| Weight | 21.34 | 4.51 | 0.94 |
| Neuron | 21.09 | 4.37 | 0.50 |

Table 11: Full results of ablation on selection strategy.

| | Aircraft | | | Birdsnap | | | Cars | | | CIFAR100 | | | CUB | | | GTSRB | | |
|---|---|---|---|---|---|---|---|---|---|---|---|---|---|---|---|---|---|---|
| | Acc. | F. | C. | Acc. | F. | C. | Acc. | F. | C. | Acc. | F. | C. | Acc. | F. | C. | Acc. | F. | C. |
| Weight | 44.43 | 14.42 | 63.48 | 55.35 | 12.78 | 61.94 | 77.51 | 3.26 | 63.42 | 83.99 | -0.39 | 61.38 | 71.51 | 4.84 | 62.87 | 94.25 | -7.87 | 62.55 |
| Neuron | 44.13 | 14.02 | 63.60 | 55.32 | 12.66 | 62.77 | 77.46 | 3.33 | 63.62 | 83.98 | -0.79 | 61.84 | 71.14 | 5.16 | 63.23 | 93.54 | -8.15 | 63.22 |

**Selection Rate.** Table 12 shows the full results of variants of our methods in the selection rate. Our main results select the top 10% elements localized layer. We compare to the baselines where the top 1% or the top 50% are selected for update. All other configurations are kept the same.

Table 12: Full results of ablation on selection rate.

| | Aircraft | | | Birdsnap | | | Cars | | | CIFAR100 | | | CUB | | | GTSRB | | |
|---|---|---|---|---|---|---|---|---|---|---|---|---|---|---|---|---|---|---|
| | Acc. | F. | C. | Acc. | F. | C. | Acc. | F. | C. | Acc. | F. | C. | Acc. | F. | C. | Acc. | F. | C. |
| 0.01 | 37.64 | 11.45 | 63.54 | 53.49 | 9.87 | 62.25 | 74.62 | 2.20 | 63.40 | 83.79 | -1.64 | 60.91 | 66.79 | 4.39 | 63.01 | 88.89 | -7.71 | 61.53 |
| 0.10 | 44.43 | 14.42 | 63.48 | 55.35 | 12.78 | 61.94 | 77.51 | 3.26 | 63.42 | 83.99 | -0.39 | 61.38 | 71.51 | 4.84 | 62.87 | 94.25 | -7.87 | 62.55 |
| 0.50 | 46.73 | 20.74 | 63.56 | 53.96 | 17.98 | 61.72 | 77.64 | 6.12 | 63.48 | 83.47 | 1.51 | 61.53 | 71.89 | 8.06 | 62.85 | 95.74 | -7.85 | 62.43 |

**Buffer Size.** Table 13 shows the full results of buffer size ablation. We study buffer sizes of 1%, 2% and 4% of the total dataset size.

Table 13: Full results of ablation on the buffer size.

| | Buffer Size | Aircraft | | | Birdsnap | | | Cars | | | CIFAR100 | | | CUB | | | GTSRB | | |
|---|---|---|---|---|---|---|---|---|---|---|---|---|---|---|---|---|---|---|---|
| | | Acc. | F. | C. | Acc. | F. | C. | Acc. | F. | C. | Acc. | F. | C. | Acc. | F. | C. | Acc. | F. | C. |
| ER | 1% | 27.76 | 44.49 | 49.22 | 43.76 | 33.59 | 55.90 | 61.22 | 21.19 | 55.34 | 73.70 | 13.34 | 40.14 | 53.39 | 25.26 | 48.81 | 93.03 | -4.22 | 16.79 |
| ER | 2% | 33.42 | 41.37 | 49.74 | 49.96 | 29.20 | 56.35 | 62.83 | 21.45 | 57.35 | 78.72 | 7.94 | 41.74 | 57.90 | 22.84 | 50.83 | 95.64 | -6.68 | 15.82 |
| ER | 4% | 41.42 | 31.48 | 50.41 | 56.22 | 21.63 | 56.72 | 69.08 | 16.42 | 58.07 | 82.86 | 3.41 | 42.10 | 64.07 | 17.72 | 51.30 | 96.28 | -7.40 | 17.34 |
| SPU | 1% | 37.82 | 21.96 | 63.56 | 47.54 | 23.61 | 61.65 | 73.68 | 6.84 | 63.36 | 80.44 | 4.87 | 61.49 | 66.06 | 9.32 | 62.47 | 90.55 | -4.93 | 62.76 |
| SPU | 2% | 40.65 | 20.31 | 63.44 | 51.33 | 18.97 | 61.90 | 75.00 | 6.17 | 63.41 | 82.45 | 2.03 | 61.36 | 68.39 | 8.42 | 62.81 | 92.99 | -7.04 | 62.63 |
| SPU | 4% | 44.43 | 14.42 | 63.48 | 55.35 | 12.78 | 61.94 | 77.51 | 3.26 | 63.42 | 83.99 | -0.39 | 61.38 | 71.51 | 4.84 | 62.87 | 94.25 | -7.87 | 62.55 |

**Number of samples to compute gradient approximation.** In Equation 2, we accumulate the gradients of $N_t'$ samples to approximate the importance. Here we ablate the effect of accumulating gradients with one batch, 128 data point, 25% 50%, and 100% of the current set. With few samples to approximate the scoring function, the computational efficiency will be enhanced. With more samples, the accuracy is slightly increased, with also slight decrease of forgetting. Our algorithm is robust to all different configurations in general. Full results are shown in Table 14

Table 15: Ablation study on gradient accumulation samples

| | Avg. Acc. Impr. | Avg. F. | Avg. C. Drop |
|---|---|---|---|
| one batch | 21.24 | 4.64 | 0.94 |
| 25% | 21.34 | 4.51 | 0.94 |
| 50% | 21.35 | 4.46 | 0.94 |
| 100% | 21.37 | 4.43 | 0.96 |

Table 14: Full results of ablation on the number of samples to compute the gradient approximation.

| | Aircraft | | | Birdsnap | | | Cars | | | CIFAR100 | | | CUB | | | GTSRB | | |
|---|---|---|---|---|---|---|---|---|---|---|---|---|---|---|---|---|---|---|
| | Acc. | F. | C. | Acc. | F. | C. | Acc. | F. | C. | Acc. | F. | C. | Acc. | F. | C. | Acc. | F. | C. |
| one batch | 44.42 | 14.40 | 63.50 | 55.02 | 13.33 | 62.04 | 77.41 | 3.36 | 63.40 | 83.99 | -0.38 | 61.36 | 71.48 | 4.90 | 62.86 | 94.14 | -7.79 | 62.52 |
| 0.25 | 44.43 | 14.42 | 63.48 | 55.35 | 12.78 | 61.94 | 77.51 | 3.26 | 63.42 | 83.99 | -0.39 | 61.38 | 71.51 | 4.84 | 62.87 | 94.25 | -7.87 | 62.55 |
| 0.50 | 44.33 | 14.48 | 63.48 | 55.31 | 12.73 | 61.88 | 77.54 | 3.16 | 63.44 | 84.03 | -0.41 | 61.37 | 71.67 | 4.63 | 62.87 | 94.24 | -7.82 | 62.58 |
| 1.00 | 44.40 | 14.40 | 63.47 | 55.28 | 12.61 | 61.85 | 77.61 | 3.12 | 63.44 | 84.05 | -0.40 | 61.35 | 71.66 | 4.64 | 62.87 | 94.27 | -7.81 | 62.58 |

