# OpenReview forum: "Overcoming Generic Knowledge Loss with Selective Parameter Update"
_ICLR.cc/2024/Conference — ICLR 2024 Conference Withdrawn Submission_

### Official Review · Reviewer_J6w2 · 2023-10-29

**Soundness:** 2 fair
**Presentation:** 3 good
**Contribution:** 3 good
**Rating:** 5
**Confidence:** 5

**Summary:**

This paper introduces a method that selectively updates pre-trained model parameters when continually training on downstream tasks, aiming at preserving the generic knowledge in the foundation model. Two strategies (Gradient-Based and Learnable) are proposed to identify the sparse parameters. The experiments are focused on the CLIP model and evaluated on six downstream computer vision tasks.

**Strengths:**

1. The presentation of this paper is clear and easy to follow.
2. Preserving generic knowledge in foundation models during continual learning and providing quantitative assessment for generic knowledge loss is meaningful.
3. The results based on the CLIP model are good.

**Weaknesses:**

1. The quantitative evaluation of generic knowledge loss can only be applied to vision-language models. Specifically, such control set Accuracy (C.) is represented by the zero-shot performance differences between the tuned and frozen pre-trained models, which cannot be obtained for pre-trained unimodal models.

2. The reasons for **only** updating the (first) MLP layer in each transformer block are not sufficient. This paper ignores the discussion of other components, such as the attention block and normalization layers. The previous work [A] mentioned in this paper does not suggest that MLP layers are more important than other layers.

    Besides, as for fine-tuning ViT, [B] suggested that "fine-tuning attention is all you need" since tuning attention weights are more efficient than tuning MLP parameters while reaching similar performance. It is supposed to include such discussion since this paper claims to find an efficient way for tuning foundation models.

    On the other hand, the generalization of the Parameter Selection operation may be limited if it cannot automatically select parameters when all parameters of the transformer are included without the manual Localization operation.

4. Missing comparisons and discussion. This work compares two baselines, FLYP [C] and ZSCL [D], in the main text (and two prompt-based methods in the appendix). Note that FLYP was not proposed for the continual setting. Therefore, this paper combines it with four continual learning methods, which makes sense. However, a closely related (but not mentioned) work [E] indicated that the pre-trained foundation model can naturally preserve generic knowledge even when tuning all parameters continually. It is supposed to include discussion and comparisons with [E] as a baseline to support this work.

5. (Minor) Lots of reference links are invalid, which makes it hard to read.

References.
[A] Transformer Feed-Forward Layers Are Key-Value Memories. EMNLP 2021.
[B] Three things everyone should know about Vision Transformers. ECCV 2022.
[C] Finetune like you pretrain: Improved finetuning of zero-shot vision models. CVPR 2023.
[D] Preventing Zero-Shot Transfer Degradation in Continual Learning of Vision-Language Models. ICCV 2023.
[E] SLCA: Slow Learner with Classifier Alignment for Continual Learning on a Pre-trained Model. ICCV 2023.

**Questions:**

See Weaknesses part for details.

---

### Official Review · Reviewer_KY5L · 2023-10-31

**Soundness:** 2 fair
**Presentation:** 3 good
**Contribution:** 2 fair
**Rating:** 3
**Confidence:** 3

**Summary:**

This paper introduces a novel approach to updating foundation models by focusing updates on relevant parameters for the specific task. The method achieves improved accuracy on new tasks while maintaining the model's transferability and generalizability. Experimental results are widely validated and demonstrate up to 7% accuracy improvement on newly learned tasks with minimal impact on overall accuracy, to show the effectiveness of the proposed method.

**Strengths:**

1. Clear paper writing.

2. The investigated task is interesting.

3. The direction of the idea seems to be plausible.

**Weaknesses:**

This paper's main flaw is insufficient comparison and elaborations regarding closely related methods.

Please refer to my questions below for more details.

**Questions:**

1. This work should prove its generality and localization strategy on different transformer architectures. However, relying solely on CLIP as the backbone, which already excels in zero-shot learning, limits the ability to prove the conclusions. For class incremental tasks, there are numerous related works in the community, such as VIT.

2. The selection of the MLP layer for localization in this work is only based on intuitive inference and assumptions. What is the difference between the learnable parameters in normalization, embedding, and attention layers? For example, lots of works choose to update the parameters in the attention layers.

3. This work is similar to some parameter-efficient fine-tuning methods, like LoRA and Adapter tuning. It needs more comparisons. And what are the advantages of the selective parameters compared with the low-rank?

4. In the context of gradient-based selection strategies, it is possible that the selected parameters are fundamental and crucial for the model's general ability, thereby being sensitive to all tasks.

---

### Official Review · Reviewer_RovQ · 2023-10-31

**Soundness:** 2 fair
**Presentation:** 2 fair
**Contribution:** 2 fair
**Rating:** 3
**Confidence:** 3

**Summary:**

This work looks at the problem of updating a pretrained model while keeping performance on the pretrained task. To do so, it proposes SPU, which is based on 3 parts:  (1) limit updates to only the first layer of the MLP in a transformer architecture (2) use the loss on a small set of examples to select a small set of parameters of the selected layers (3) train the sparse selected parameters.

It evaluates the proposed method in a single model (CLIP-ViT-B) on 6 tasks. It compares against methods which tune all parameters.

**Strengths:**

There seems to be a strong effect on limiting the parameters to be updated and maintaining performance on pretrained tasks (as shown in table 5). This work exploits that to present SPU as performing better than alternatives that finetune all parameters.

**Weaknesses:**

Lack of baselines: Given that the number of parameters being tuned has a significant impact on new task and pretrained task performance, I think there is a lack of baselines around this to justify the presented method. In particular how do existing PEFT methods such as Adapters, LoRA, KAdaptation perform vs SPU.

Lack of clarity: SPU is presented in a rather convoluted way. IIUC SPU is a rather simple method and that should be a plus if presented as so. Unfortunately in my opinion that is not how the text presents it. E.g. no need to write the fixed decision of only using the first MLP layer as a phase of the algorithm, similar I think the text on the selection of parameters is a rather indirect way to say it selects the parameters with the highest absolute loss derivative computed on a sample of examples.

Incomplete ablation: There is an ablation indicating that the first layer is better to update the parameters, however there appears to be no sweep of other hyper parameters such as the percentage of values being tuned. Similarly when comparing gradient vs learnable score it seems both have different tradeoffs of accuracy/forgetting and sweeping other parameters they could reach different tradeoffs.

Train cost efficiency: The gains on batch time are suspiciously high, which could be true in a specific hardware/configuration/batch size/communication/memory limitations/etc but maybe not representative of performance gains one would observe in a well tuned baseline. Without further information on where the gains come from it is hard to judge the claimed values.

**Questions:**

a) How do recent parameter efficient tuning methods compare against SPU?

b) When plotting ‘accuracy’ and ‘control accuracy’ is SPU pareto optimal with existing methods? If not, then when does it works better than existing methods (e.g. sweeping LoRA rank vs sweeping percentage of SPU paramaters)?

c) Where do the train cost efficient gains from the reduced number of parameters come from?

---

### Official Review · Reviewer_N81h · 2023-11-01

**Soundness:** 3 good
**Presentation:** 3 good
**Contribution:** 2 fair
**Rating:** 5
**Confidence:** 4

**Summary:**

To make the foundation models applicable to new tasks while retaining the generic knowledge gained through pre-training, this paper proposes a method for fine-tuning the model. The authors first identify the layers that need to update parameters, then select a small subset of parameters that need to be updated by designing a scoring function, and finally freeze the other parameters and train the model on a new dataset. Experiments show that compared with the latest fine-tuning and incremental learning methods, the method proposed in this paper achieves high accuracy in the new task without significant performance degradation on the original dataset.

**Strengths:**

1)New ideas: This paper proposes that only a few parameters that are sparse but most important for new tasks need to be updated.

2)New technologies: The authors selected the parameters most highly relevant to the new task by designing a learnable gradient-based scoring function.

3)Better experimental results: This method is superior to other latest methods in accuracy and forgetting rate of image classification, indicating that the knowledge obtained from model pre-training is retained in continuous learning.

**Weaknesses:**

1）The paper lacks sufficient theoretical support and generalization ability to explain how to identify the specific layers that need parameter updates. It is not sufficient to argue that only the parameters in the MLP layer need to be updated by pointing out that the MLP layer has the ability of pattern detection. The paper's final choice to update only the first MLP layer is determined by designing a permutation experiment. Imagine that when the number of layers to be selected becomes larger, determining the final position requires a large number of experiments. This indicates that there is no clear theoretical guidance for locating specific layers. When dealing with non-transformer-based models, the paper does not specify how to locate the specific layers that need to modify their parameters.



2）Neither of the two scoring functions designed in this paper stands out significantly. The paper points out that learnable scoring functions will improve gradient-based methods for a range of problems, at the cost of more computation. However, according to the experimental results in Table 3, it can be seen that the learnable function only has some advantages in the accuracy of the control set, and it can be seen from the data that this advantage is very small. This does not fully demonstrate that it eliminates some of the drawbacks of gradient-based scoring functions.

3) On some key issues, the paper lacks in-depth analysis and experiments. First, the paper posits that parameters with a more substantial impact on gradients hold greater significance for new tasks, without further analyzing what these parameters represent and their specific relationship with new tasks and general knowledge. Second, other than localization and sparsity, the paper does not analyze much of how or why their method retains the generic knowledge but focuses more on the relationship of the model to new tasks. Third, in addition to the CLIP model, I suggest that experiments can be conducted on more foundation models, and the control set can not be limited to ImageNet, but select multiple representative datasets to reflect the experimental results.

**Questions:**

Please refer to the weakness section above.